# Cell-free RNA reveals host and microbial correlates of broadly neutralizing antibody development against HIV

Mark Kowarsky[1], Mercedes Dalman[2,3], Mira N. Moufarrej[4], Jennifer Okamoto[5], Yike Xie[2,3], Norma F. Neff[5], Salim S. Abdool Karim[6,7], Nigel Garrett[6,8,9], Penny L. Moore[6,10,11,12]*, Joan Camunas-Soler [2,3,13]*, Stephen R. Quake[4,5]*

1 Department of Physics, Stanford University, Stanford, California, United States of America, 2 Department of Medical Biochemistry and Cell Biology, Institute of Biomedicine, University of Gothenburg, Gothenburg, Sweden, 3 Wallenberg Centre for Molecular and Translational Medicine, Sahlgrenska Academy, University of Gothenburg, Gothenburg, Sweden, 4 Department of Bioengineering, Stanford University, Stanford, California, United States of America, 5 Chan Zuckerberg Biohub, San Francisco, California, United States of America, 6 Centre for the AIDS Programme of Research in South Africa (CAPRISA), University of KwaZulu Natal, Durban, South Africa, 7 Department of Epidemiology, Columbia University, New York, New York, United States of America, 8 Desmond Tutu HIV Centre, University of Cape Town, Cape Town, South Africa, 9 Discipline of Public Health Medicine, School of Nursing and Public Health, University of KwaZulu-Natal, Durban, South Africa, 10 SA MRC Antibody Immunity Research Unit, Faculty of Health Sciences, University of the Witwatersrand, Johannesburg, South Africa, 11 Centre for HIV and STIs, National Institute for Communicable Diseases (NICD) of the National Health Laboratory Service (NHLS), Johannesburg, South Africa, 12 Wits Infectious Diseases and Oncology Research Institute, Faculty of Health Sciences, University of the Witwatersrand, Johannesburg, South Africa, 13 Science for Life Laboratory, Institute of Biomedicine, University of Gothenburg, Gothenburg, Sweden

* steve@quake-lab.org (SRQ); pennym@nicd.ac.za (PLM); joan.camunas@gu.se (JC-S)

## Abstract

A small number of people living with HIV (PLWH) develop broadly neutralizing antibodies (bNAbs) targeting multiple HIV strains. Although several viral and immune factors contribute to bNAb development, the genetic and environmental factors driving this response remain largely unknown. We performed combined cell-free DNA (cfDNA) and cell-free RNA (cfRNA) sequencing in 42 plasma samples from a longitudinal cohort of 14 PLWH (7 who develop bNAbs and 7 matched controls). This approach enabled us to non-invasively monitor the host transcriptome, viral genetic variation, and microbiome composition during HIV infection, and to identify molecular correlates of bNAb development. We find that development of bNAbs is associated with a transcriptomic signature of early immune activation characterized by elevated levels of MHC class I antigen presentation genes. This signature is independent of viral load or CD4 count and declines over time. In addition to host features, we recovered sufficient viral reads to reconstruct HIV consensus sequences, supporting the utility of cfRNA for viral genotyping. Finally, we also identified an enrichment of several microbial taxa in bNAb producers and increased levels of GB virus C (GBV-C), a non-pathogenic lymphotropic virus. Our findings suggest a distinct early immune activation profile in PLWH who develop bNAbs. More broadly, we show that

**Data availability statement:** The datasets generated and analyzed in the study are available in the NCBI Gene Expression Omnibus (GEO) and Sequence Read Archive (SRA) with accession number GSE313190. The code to reproduce the analyses in this study is publicly available at https://github.com/CamunasLab/cfHIV.

**Funding:** This work was supported by the funding from the Chan Zuckerberg Biohub to S.R.Q, the Bill & Melinda Gates Foundation (OPP1113682) and the Global Health Vaccine Accelerator Platform (GH-VAP-SI-ID-14) to S.R.Q and P.L.M. P.L.M. is supported by South African Medical Research Council (MRC) SHIP program and the South African Research Chairs Initiative of the Department of Science and Technology and the NRF (Grant No 98341). J.C.-S. is supported by the Knut and Alice Wallenberg Foundation (Wallenberg Molecular Medicine Fellow, the Swedish Research Council (2021-05109) and the Erling Perssons Stiftelse (Swedish Foundations' Starting Grant). M.N.M. was supported by the Stanford Bio-X Bowes Fellowship. The funders had no role in study design, data collection and analysis, decision to publish, or preparation of the manuscript.

**Competing interests:** The authors have declared that no competing interests exist.

combined cfDNA/cfRNA sequencing can reveal relationships between a protective immunogenic response to HIV infection, the host immune system, and microbiome, highlighting its potential for biomarker discovery in future vaccine and therapeutic studies.

## Author summary

A subset of people living with HIV develop broadly neutralizing antibodies (bNAbs), a phenomenon that remains poorly understood but relevant to vaccine development. We applied combined cell-free RNA and DNA sequencing to 42 longitudinal plasma samples (7 bNAb producers, 7 controls) to monitor the host transcriptome, viral variation, and microbiome composition during bNAb development. Early in infection, bNAb producers showed an MHC class I-linked immune activation signature that declined over time, alongside changes in commensal microbial sequences and increased levels of GB virus C in cfRNA. These findings support the use of cfDNA/cfRNA sequencing for biomarker discovery in HIV research.

## Introduction

HIV has infected over 36 million people worldwide, reaching epidemic proportions in Eastern and Southern Africa with an estimated adult prevalence close to 7% [1]. Antiretroviral therapy (ART) has greatly improved outcomes, allowing HIV to be a manageable condition throughout life. However, global access to ART is difficult to achieve and also requires lifelong daily adherence to a pill taking regime. Moreover, despite effective treatment, transmission of HIV remains common in vulnerable populations, and people living with HIV (PLWH) are at increased risk of aging-related and chronic diseases [2].

Although virtually all PLWH develop strain-specific autologous responses [3–6], few develop broadly neutralizing antibodies (bNAbs), which can neutralize multiple strains of the virus [7], and when administered passively can prevent acquisition of susceptible HIV strains and suppress viral replication in PLWH [8]. There is great interest in understanding the drivers of bNAbs, as this could inform the development of an effective vaccine. Efforts to design an effective HIV vaccine have increasingly focused on strategies to induce bNAbs by guiding B-cell affinity maturation, with studies demonstrating promising immunization approaches through the rational design of immunogens [9–11]. Despite these advances, which specific cellular processes give rise to bNAbs during infection remains incompletely described. So far, there is evidence that viral and host genetic factors might play a role [12–16], and that a combination of high viral titres and reduced CD4 counts early in infection is associated with the development of bNAbs in some individuals [17,18]. Additionally, changes in frequency of certain immune subpopulations, such as T follicular helper cells (Tfh)

might also influence affinity maturation towards bNAb production [19–23]. Moreover, there is some evidence that bNAbs might arise from pre-existing memory B cells initially stimulated by bacterial antigens -likely from the gut microbiome- which later cross-react with the HIV glycan shield and develop neutralizing capabilities [24,25]. Although microbial antigen cross-reactivity has also been shown to lead to non-protective immune responses in HIV vaccine trials [26], a better understanding of the interplay between the host immune response, microbiome and a protective bNAb response could aid in vaccine development.

Sequencing of plasma cell-free DNA (cfDNA) has become an established tool in clinical medicine, with applications ranging from non-invasive prenatal testing and oncology to organ transplant monitoring [27,28]. In contrast, plasma cell-free RNA (cfRNA) provides an emerging complementary approach to measure dynamic changes in the host transcriptome [29]. Because transcripts released into circulation can originate from many different cell types and tissues [30], cfRNA enables non-invasive monitoring of systemic physiological and immune states [31,32]. To date, cfRNA sequencing has been used in diverse biomedical settings, including the monitoring of placental and pregnancy complications for maternal-fetal health [33–36], to follow bone marrow reconstitution after stem cell transplantation [37], and to characterize host immune responses to tuberculosis infection [38]. However, integrative analyses that capture host, viral, and microbial nucleic acids simultaneously remain uncommon, and cfRNA sequencing approaches have been rarely applied to monitor retroviral infections such as HIV. Here, we conducted a pilot study to investigate whether combined cfDNA and cfRNA sequencing of plasma could reveal molecular features associated with the development of bNAbs in a longitudinal cohort of PLWH.

## Results

### Study design, sequencing overview and viral analysis

We analyzed 42 plasma samples from 14 PLWH from the CAPRISA studies, of whom 7 developed bNAbs and 7 did not. For each participant, plasma was collected at approximately 6 months, 1 year, and 3 years post-HIV acquisition and prior to ART (Fig 1A). At the 6-month time point, both groups had comparable CD4 counts and viral loads (S1 Fig). The extent of bNAb development was assessed by measuring plasma neutralization breadth against a panel of 18 Env-pseudotyped HIV viruses, summarized as percentage breadth, a continuous measure of bNAb development (Fig 1B and S1 Table and Methods).

We performed combined cfDNA and cfRNA sequencing in each of the 42 samples (Methods). After quality control and filtering, we obtained enough reads to sample the host transcriptome (80% of total reads for cfRNA) and microbiome (0.2% and 20% reads of cfDNA and cfRNA respectively) (S2 Fig). Sequencing depth before and after sample QC processing (cleaning) is similar for both cfDNA and cfRNA, with comparable metrics between bNAb producers and non-producers (S3 Fig). Similarly, cell-type deconvolution of the host transcriptome showed comparable contributions of the most abundant cell types (e.g., red blood cells, immune cells, platelets) across both groups (S4 Fig). We detected HIV reads in the cfRNA assay for all samples, with coverage sufficient to genotype the majority strain in 38 out of 42 samples (S5A and S5B Fig). For high coverage samples, the nucleotide identity of HIV clustered within individuals based on previous single-genome amplified envelope sequences (Fig 2A) except for one mislabeled sample that was excluded from further analyses (Methods).

We validated the accuracy of cfRNA-derived consensus sequences by comparing them to previously published envelope sequences from the same CAPRISA participants (S2 Table), observing consistent clustering and strain concordance across time points and sequencing methods (Figs 2B and S5C). By comparing differences in consensus sequences within individuals (S5D Fig), we found that most mutations occur in the *env* gene, which encodes the surface protein responsible for host cell binding. The Env protein is the sole target of bNAbs and is known to be the most rapidly evolving part of the HIV genome [39–41]. Using cfRNA we estimated that the dominant viral strain in each individual accumulates approximately 10 mutations per year (S5E Fig), in line with other estimates from virion or PBMC-derived studies [42,43].

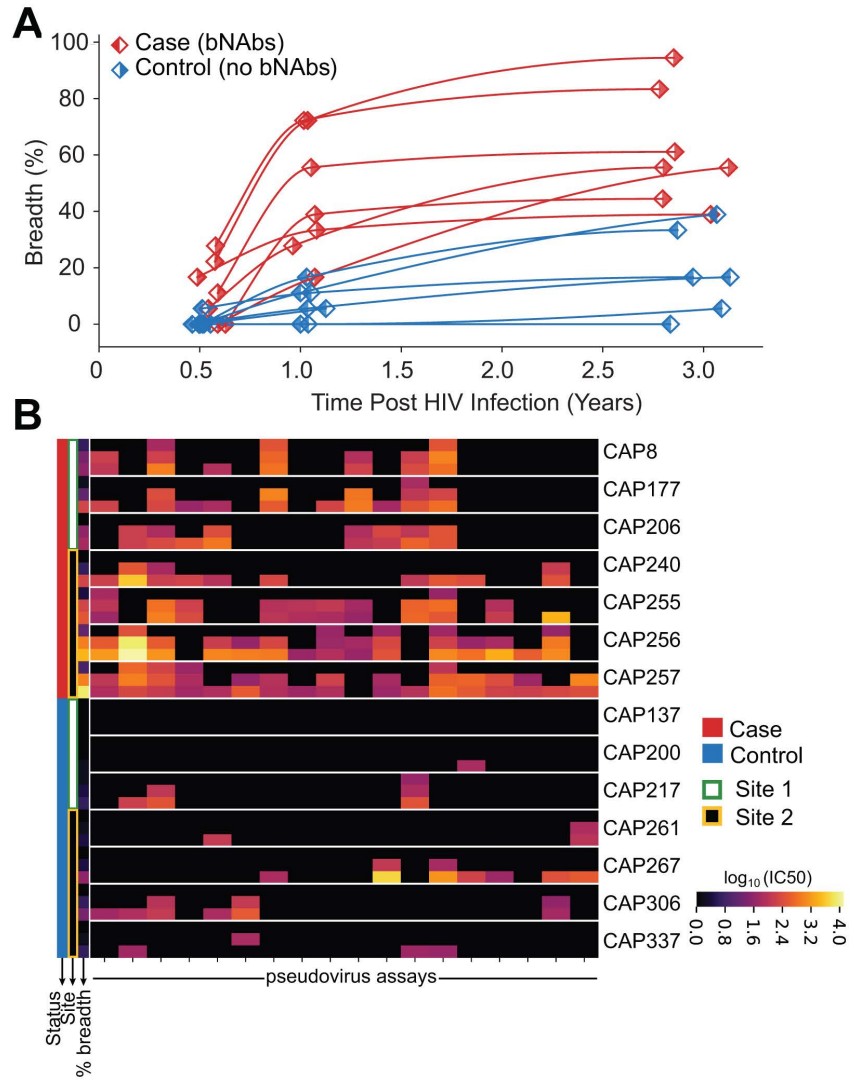

**Fig 1. Study design and development of neutralization breadth. A)** Longitudinal trajectories of neutralization breadth in bNAb producers (red) and controls (blue). Time zero corresponds to the estimated HIV infection point, with breadth defined as the ability of plasma antibodies to neutralize a cross-clade panel of 18 env-pseudotyped viruses (S2 Table). **B)** Heatmap of serum neutralization titers (IC50) against the 18-virus panel, grouped by individual, with time points sorted from first (top) to last (bottom). Colors represent log10-transformed IC50 neutralization titers, with higher values (yellow) indicating stronger neutralization potency and black indicating values below the limit of detection.

Altogether, these results show that cfRNA is a quantitative tool to perform genotype analysis of viral pathogens and to track strain-specific mutational dynamics over time.

Analysis of read lengths and mapping start and end positions indicates that HIV-derived reads in plasma cfRNA (cfHIV) predominantly consist of short RNA fragments (200–500 bases) and do not show systematic positional biases along the HIV genome (S6 Fig). This is consistent with cfHIV being largely derived from fragmented cell-free viral RNA rather than libraries dominated by intact full-length viral genomes. In addition, to assess the quality of HIV genome recovery from cfRNA, we performed a subsampling analysis showing that approximately 50,000 cfHIV reads are sufficient to recover >90% of the genome at ≥50X depth (S7 Fig). In this cohort, this level of coverage was achieved for most samples (83%).

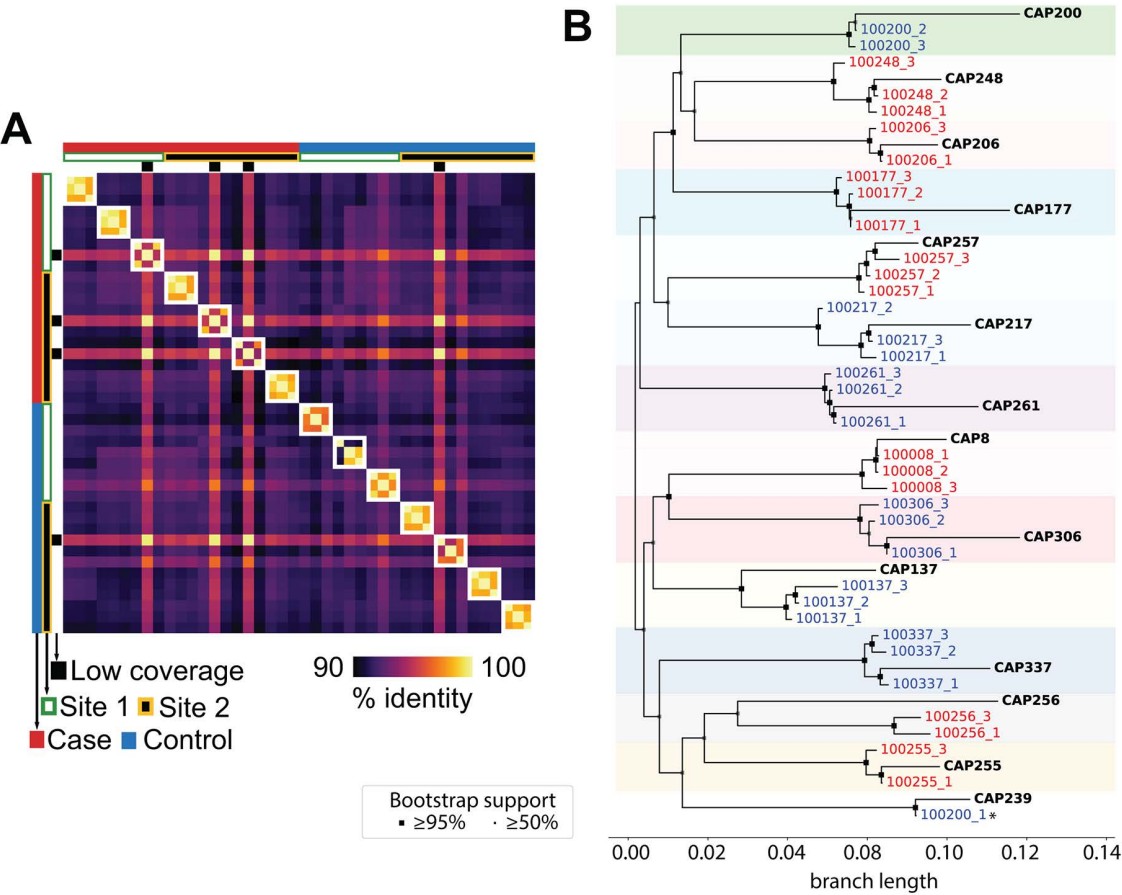

**Fig 2. HIV consensus sequences from cfRNA. A)** Genomic similarity matrix of consensus HIV sequences obtained from cfRNA, showing clustering within individuals. **B)** Maximum-likelihood phylogeny of cfRNA-derived HIV consensus sequences (colored labels) with previously published envelope sequences (bold, CAPRISA). Shaded boxes highlight individual-level clustering. With the exception of one sample, cfRNA consensus sequences consistently cluster with the corresponding envelope sequences.

## Host transcriptome correlates of cfHIV and clinical parameters

To characterize host responses to HIV infection, we measured the correlation between the host transcriptome and the total abundance of cfHIV, as well as with clinical parameters such as viral load and CD4 counts. Spearman correlation of cfHIV, viral load and CD4 counts to the human transcriptome revealed several genes associated with these measurements (Figs 3A and S8A and S8B). Analysis of genes significantly correlated to cfHIV revealed 19 genes typically associated with host immune responses to viral infection (FDR < 0.05), whereas only 1 for each of viral load and CD4 count, respectively (Fig 3B). CD4 count levels showed an association with *IL4*, a cytokine inducing differentiation of naive T helper cells. Although the relation between cytokine profile and HIV progression is controversial, increased secretion of IL4 in relation to IFN-gamma is a known hallmark of switching from Th1 to Th2 lymphokine secretion in PLWH [44]. Our observation of IL4 being increased in low CD4 samples is compatible with an increase of IL-4 levels upon disease progression and T helper cell decrease. Top correlates of cfHIV include: *CCL5*, also known as RANTES, an HIV suppression factor released by CD8⁺ T cells, *TFRC* (CD71), a marker of T-cell activation, and *CCL2*, a chemokine that attracts immune cells to sites of infection (Figs 3C and S8C). These genes do not show a differential behaviour between bNAb producers

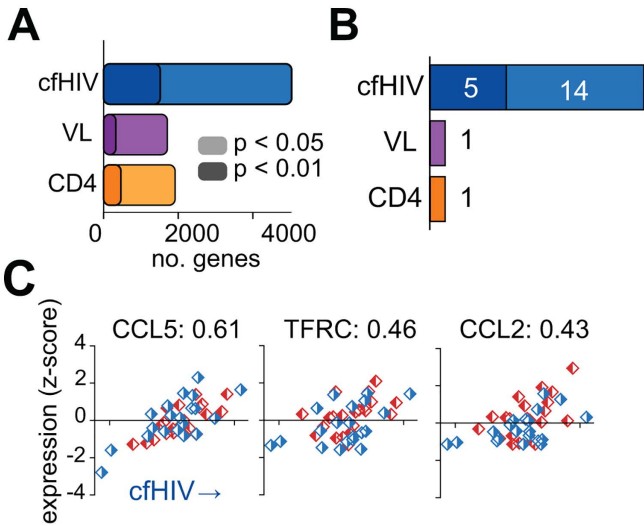

**Fig 3. Correlation between cfHIV levels and host transcriptome. A)** Number of genes significantly correlated with cfHIV counts, viral load (VL) and CD4 counts (Spearman's rank correlation, see Methods). **B)** Subset of genes from panel A previously described as involved in host immune responses to viral infection. **C)** Example correlations between cfHIV counts and three HIV-associated genes. Spearman correlation coefficients indicated. No partitioning is observed between bNAb producers (red) and controls (blue).

and controls, indicating that they represent shared responses to HIV infection. This shows that cfRNA sequencing captures shared host transcriptional responses associated with peripheral viral burden during HIV infection.

## Development of bNAbs is associated with a transcriptional phenotype of immune activation

We next compared the transcriptome of bNAb producers and non-producers. We performed differential gene abundance analysis finding 256 genes elevated in bNAbs producers (FDR < 0.05, fold-change>1, Fig 4A, Methods). This variation could not be attributed to environmental factors such as site of collection, that showed little to no correlation to these transcriptomic signatures. Remarkably, 52% (n = 131) of elevated genes in bNAb producers are immune-related (primarily associated with adaptive immunity), compared with only 14% (n = 39) of the 281 decreased genes. Among the elevated genes, 25 of them are associated with production of RANTES (CCL5), a chemokine involved in homing and migration of effector and memory T cells. Although CCL5 itself is not enriched in bNAb producers in our dataset, elevated RANTES, along with other inflammatory markers, have previously been linked to the development of bNAbs in spontaneous controllers of HIV [45].

A pathway analysis confirmed these results and further revealed an enrichment of molecular functions related to an immunologic response in bNAb producers (S9A Fig). Terms such as receptor activity, IgG and peptide antigen binding, cytokine signaling and NF-kB signaling were enriched in bNAb positive samples. Analyses of the genes decreased in the bNAb producers showed lower enrichment scores and did not point to the existence of other host immunoregulatory effects (S9B Fig). Overall, host cfRNA sequencing provides evidence for an increased immune activation in participants who developed bNAbs, despite having similar CD4 counts and viral loads to non-bNAb producers.

To investigate this immune activation in more detail, we performed pseudotime analysis on the two groups, using plasma neutralization breadth to define the temporal progression (Fig 4B, Methods). Pseudotime is a trajectory-inference approach that orders samples by relative biological state rather than chronological time, allowing comparison across individuals at different stages of bNAb development. We found that the immunologic gene signature is elevated only early in breadth development in bNAb producers (when only a limited number of viral strains are neutralized) and later converges

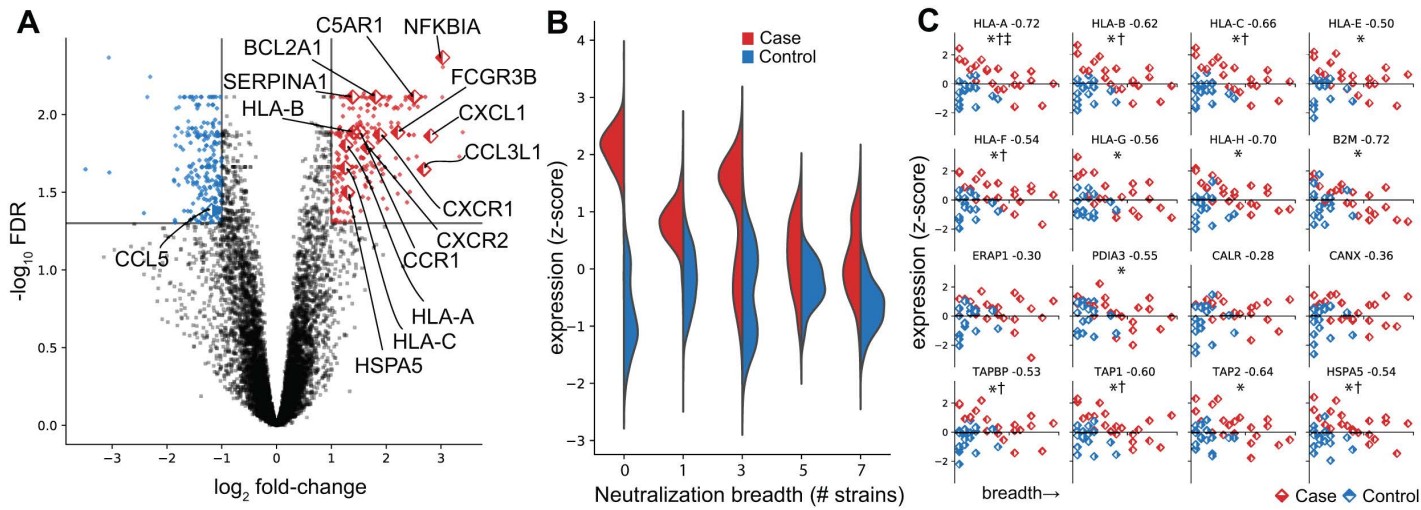

**Fig 4. Gene level differences in cfRNA between bNAb producers and controls. A)** Volcano plot of differentially abundant genes. Genes with a fold change > 2 and FDR < 0.05 are highlighted (red = increased, blue = decreased in bNAb producers). Labelled points are genes involved in host immune responses to viral infection. **B)** Violin plots of the distribution of elevated genes in bNAb producers, stratified by group and neutralization breadth (number of viral strains neutralized), used here as a proxy for progression along bNAb development. **C)** Genes in the MHC class I antigen presentation pathway. Most show negative correlations with breadth, and some are also differentially abundant between bNAb producers and controls. Correlations with breadth were assessed using Spearman's rank correlation (Spearman's ρ indicated next to each gene, see Methods). * correlated with breadth at FDR < 0.05, † differentially abundant between bNAb producers and controls at FDR < 0.05, ‡ correlated with cfHIV at FDR < 0.05.

to levels comparable to non-bNAb producers as neutralization breadth increases. This aligns with previous studies showing that factors such as CD4 levels inversely correlate with bNAb development only early in infection [4]. However, in our study, both groups had similar CD4 counts at the first time point. Focusing on the genes significantly correlated with breadth in both groups (S9C and S9D Fig, FDR < 0.05) revealed almost the entirety of the MHC class I antigen presentation pathway was anticorrelated with the development of breadth (i.e., pseudotime) and sometimes differentially altered or correlated with the level of cfHIV measured (Fig 4C). Hence, MHC class I antigen genes are highly elevated early in infection in participants who go on to develop bNAbs, and their abundance decrease as breadth develops, becoming comparable to that of participants who do not develop bNAbs. MHC class I antigen presentation is present in almost all cells of the body, presents antigens to be killed by cytotoxic T cells (CD8) but is not generally associated with antigen presenting cells or the production of antibodies by B cells [4]. Taken together, these results indicate that early immune activation associated with MHC class I antigen presentation, independent of viral load or CD4 count, is a relevant factor in the development of bNAbs.

## Microbiome associated with bNAbs and the development of breadth

Development of bNAbs has been hypothesized to be triggered by antigen cross-reactivity between HIV and bacterial components present in the gut microbiome [24,25,46,47]. To test whether a differentiated microbiota could be observed in participants who developed bNAbs, we mapped all non-human reads (both cfDNA and cfRNA) to a database of bacterial and viral genomes. After discarding reads that mapped to potential reagent contaminants (S10 Fig, Methods) [48], we identified 249 taxa in cfDNA and 4366 taxa in cfRNA that could be confidently attributed to the participants microbiome. We combined data from all participants to assemble a phylogenetic tree of the human microbiome during HIV infection as measured from circulating nucleic acids (S11 Fig). This accounted for ~70% of non-human reads in both cfDNA and cfRNA, and spanned several areas of the tree of life including archaea, bacteria and both DNA and RNA viruses. Among

other features, we identified the presence of several DNA and RNA viruses such as Anelloviruses, Flaviviruses, Herpesviruses indicative of concomitant viral infections for several participants. In cfRNA, we found that ribosomal reads encompass 37% of the nonhost microbiome reads, and we also identified a moderate fraction of reads (~20%) that could not be attributed to specific species and mapped to uncultured bacteria.

We then compared the microbiome of both groups (bNAb producers/controls) finding 19 genera that were differentially abundant between both groups (FDR < 0.05, |log₂FC| > 1, Fig 5A). Among genera enriched in bNAb producers, we identified several taxa that include opportunistic pathogens and are commonly associated with mucosal surfaces and the gastrointestinal tract, including Enterobacter, Serratia and Leclercia (S12 Fig). By performing a temporal analysis, we found that the detection of these pathogens does not differ across timepoints and does not precede development of breadth (S13 Fig). Overall, and despite the relatively small sample size of this study, our data shows the existence of differences in the microbiota compositions in participants who develop bNAbs from those who do not.

Additionally, metagenomic analysis of cfRNA revealed the presence of GB virus C (GBV-C, previously known as Hepatitis G) in several participants (Fig 5B). GBV-C is a flavivirus that infects lymphocytes, has no known pathology, and has been associated with improved survival in people with HIV infection [49,50]. Given the phylogenetic relationship between GBV-C and hepatitis C virus (HCV), we also examined reads mapping to HCV. However, we did not find evidence of HCV signal, whereas reads mapping to GBV-C showed broad coverage across the viral genome (S13 Fig). In our study, all bNAb producers had non-zero levels of GBV-C (p < 0.05, one-sided Fisher exact test) and showed higher odds of elevated GBV-C levels (Fig 5B). Differential gene abundance analysis on samples with high levels of GBV-C (defined as >1/2000 nonhost reads) revealed 42 host genes that are also elevated in these participants. Among top candidates we identified several genes of the MHC class II antigen presentation complex (*HLA-DQA1*, *HLA-DQB1*, *HLA-DRB1*), an HIV suppressive factor involved in inflammatory signaling (*CCL3*), several proinflammatory cytokines (*IL6*, *EBI3*), and immunoglobulin receptors (*KIR3DL2*). Notably, increased expression of inhibitory KIRs, including KIR3DL2, has been associated with

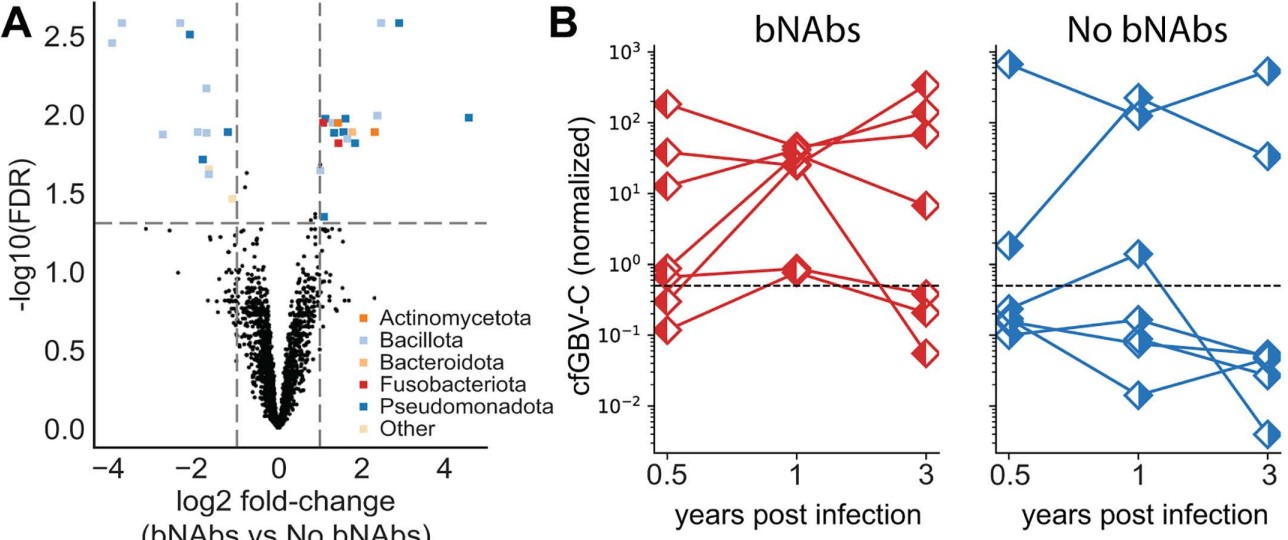

**Fig 5. Microbial differences and GBV-C enrichment in bNAb producers. A)** Volcano plot of differentially abundant microbial genera between bNAb producers and non-producers, colored by phylum (FDR < 0.05, Benjamini-Hochberg). Positive log2 fold change indicates enrichment in bNAb producers. **B)** Scatterplots showing cfGBV-C abundance in samples partitioned by time and study group (left, bNAbs producers, right, controls). The dashed line indicates the threshold used to define elevated cfGBV-C levels. Using a logistic mixed-effects model with participant as a random effect, bNAb producers showed higher odds of elevated GBV-C levels (odds ratio [OR]: 6.4, 95% CI: 2.3-17.9). In some participants, GBV-C read counts exceeded those of HIV up to 100-fold.

bNAb development, suggesting a potential link between natural killer cell maturation and the regulation of antibody production [51]. These findings suggest that high levels of GBV-C might be associated with an increased activity of antigen presenting cells.

## Discussion

In this work, we demonstrate simultaneous monitoring of viral infection and host response through sequencing of circulating nucleic acids (DNA and RNA) in plasma samples of PLWH prior to ART. We show that cfRNA can be used to characterize circulating HIV viral strains and, at the same time, to profile the host transcriptome, allowing characterization of immune activity during HIV infection. Analysis of fragment length and genome coverage suggests that HIV-derived cfRNA likely reflects fragmented viral RNA circulating in plasma rather than being dominated by intact full-length viral genomes. Subsampling analyses further shows that near-complete genome coverages (>90%) at moderate depths (≥50X) can be recovered from cfRNA, supporting the feasibility of viral genome analyses.

As cfRNA contains transcripts released from diverse tissues and cell types across the body [30], it offers a systemic view of immune processes that may be otherwise difficult to capture. We find that overall cfHIV levels—defined as HIV-derived fragments in cfRNA—correlate with abundance of host genes involved in immune responses to viral infections, including CCL5, TFRC, and CCL2, consistent with responses to peripheral viral burden. For instance, CCL5 encodes RANTES, a chemokine with HIV-suppressive activity that can block CCR5-mediated viral entry, while TFRC (CD71) and CCL2 (MCP-1) are involved in pathways commonly perturbed during infection [52–54]. However, when comparing participants who develop bNAbs with those who did not, we observed a distinct and orthogonal transcriptomic signature suggestive of an enhanced immune activation early in infection that was specific to bNAb producers. This response is independent of viral load and CD4 counts, and is enriched in MHC class I antigen presentation genes, RANTES-associated pathways (though not CCL5 itself), and other immune activation pathways.

Previous studies have highlighted the central role of lymph nodes in shaping humoral responses during HIV infection, yet their direct study remains challenging [23,55,56]. Cell-free RNA may provide a non-invasive window into immune activity occurring in tissue compartments, including lymphoid tissues where Tfh-B cell interactions drive affinity maturation. Hence, it is plausible that the immune-related cfRNA transcripts we report here reflects immune activity within lymph nodes, consistent with previous evidence that cfRNA can capture tissue-resident hematopoietic activity such as bone marrow progenitor transcripts [37]. However, we emphasize that cfRNA represents an integrated signal from multiple sources, and we cannot exclude the possibility that increased cfRNA levels are due to contributions of circulating immune cells or other tissues.

Analysis of non-human reads further allowed us to assess the microbiome and virome, revealing a reduced set of taxa whose presence or absence is associated with the development of bNAbs. Microbial sequences detected in plasma cfRNA/cfDNA most likely reflect leakage of microbial products into circulation due to barrier dysfunction and dysbiosis, a process well described in chronic HIV infection [57–60]. While gut microbiota has been speculated to increase antigen cross-reactivity, our data cannot establish a direct role in bNAb development. Surprisingly, we found that participants who developed bNAbs consistently had higher levels of GBV-C. GBV-C is a non-pathogenic lymphotropic flavivirus that has been associated with improved outcomes in PLWH [49], with some studies suggesting this may reflect an indirect rather than a causal association [50]. Mechanistically, co-infection with GBV-C has been reported to inhibit HIV replication and modulate cytokine release, co-receptor expression, and T-cell homeostasis [50,61,62]. In our study, GBV-C abundance correlated with increased abundance of proinflammatory cytokines and adaptive immune genes, raising the possibility that co-infection influences the host immune milieu in a way that could facilitate bNAb development. While causality cannot be inferred from our dataset, understanding such modulatory viral co-infections early after HIV acquisition could inform strategies to mimic beneficial immune environments in the context of HIV vaccine design.

Our study represents a small pilot investigation (42 samples, 14 PLWH) in a well-characterized cohort of young women from South Africa [63], followed from prior to HIV acquisition into acute infection and before initiation of ART. Biobanked

samples from such cohorts are an invaluable resource for studies of factors associated with the development of breadth, as earlier ARV treatment regimens are now standard of care. Although this provides a relatively homogeneous and well-controlled population for an initial discovery study, the small sample size limits generalizability and statistical power. For example, associations between viral load or CD4 counts and bNAb induction reported in other studies may not be detectable here due to the limit size of this pilot cohort. Expanded studies within the CAPRISA cohort or in other settings, and with finer temporal resolution, will be important to validate the findings reported here and their relationship to bNAb development. Despite these limitations, this work illustrates the potential of cfRNA/cfDNA sequencing as a non-invasive approach to interrogate host, viral, and microbial processes simultaneously from a single plasma sample. This approach could be of use to support biomarker discovery in vaccine studies and help identify biological contexts that favor the induction of bNAbs.

## Materials and methods

### Ethics statement

The CAPRISA 002 Acute Infection study was reviewed and approved by the research ethics committees of the University of KwaZulu-Natal (E013/04), the University of Cape Town (025/2004), and the University of the Witwatersrand (MM040202). All participants provided written informed consent for screening, enrolment and specimen storage.

### Participants

The study utilized biobanked plasma specimen of PLWH with known neutralization status enrolled in the CAPRISA cohorts. The CAPRISA 002 Acute Infection cohort was established in 2004 and has been following women from the early/acute stage of infection [63]. We included participants from two CAPRISA clinical sites, Ethekwini in Durban (site 1) and Vulindlela (site 2), and accounted for potential site-specific effects in all downstream analyses.

### Neutralization measurements

Neutralization breadth was measured using a standardized pseudovirus assay that included 18 Env-pseudotyped HIV viruses (S1 Table) [64,65]. Briefly, HIV pseudoviruses were produced by co-transfecting HEK293T cells with each HIV-1 Env plasmid and a pSG3 backbone. Serial dilutions of heat-inactivated plasma were preincubated with each pseudovirus, followed by the addition of TZM-bl cells. After 48h, infection was quantified by luciferase read-out, and titers reported as the reciprocal dilution where 50% of the virus is neutralized ($IC_{50}$). Breadth was defined as the percentage of pseudoviruses neutralized at an $IC_{50} \geq 1{:}45$.

Participants were classified as bNAb producers if they achieved ≥20% breadth within year of infection or ≥40% breadth against the 18-pseudovirus panel at any three years post-infection.

### Isolation of cfDNA and cfRNA and library preparation

Blood samples were collected in sodium citrate tubes and plasma extracted according to the standard clinical centrifugation protocol of the HIV/AIDS Network Coordination, and stored at -80 C. Each plasma sample was thawed once and split into two aliquots for cfDNA/cfRNA extraction (0.4-1 mL per aliquot). To minimize confounding effects from hemolysis and platelet-derived RNA, plasma samples were visually inspected prior to processing for signs of hemolysis (e.g., pink discoloration), which was not visible for any sample. Immediately before nucleic acid extraction, plasma samples were subjected to a final centrifugation step (1 min at 16,000 rpm) to remove any residual debris. cfDNA was extracted using the Qiagen Circulating Nucleic Acid kit. cfRNA was extracted using the Plasma/Serum Circulating RNA and Exosomal Purification kit (Norgen, cat 42800), followed by DNA digestion using Baseline-ZERO DNase (Epicentre) and then cleaned up using the RNA Clean and Concentrator-5 kit (Zymo) [32,66]. cfDNA/cfRNA libraries were prepared using the automated

Ovation SP Ultralow Library Systems (Mondrian ST) and the SMARTer Stranded Total RNA Pico Input Mammalian kit (Clontech) respectively [32,66]. A negative control (1 mL PBS, Gibco) was processed in parallel for every batch to control for potential contaminants [67].

## Sequencing methods

Paired end Illumina sequencing (2 × 75 bp) was performed on the Nextseq 500 for cfDNA and on the Novaseq 6000 for cfRNA. This resulted in a median of 69 million (cfDNA) and 109 million (cfRNA) reads per sample (S3 Fig).

## Sequencing analysis

Sequencing reads from cfRNA and cfDNA were processed using a shared initial pipeline. First, low quality bases and adapter sequences were trimmed by Trimmomatic v0.38 [68], and overlapping read pairs were merged using FLASH v1.2.11 [69]. Merged reads were then aligned to the UniVec database using bowtie2 to remove reads mapping to common vectors, linkers and control sequences [70]. These processing QC steps removed only a fraction of reads in cfRNA and cfDNA libraries from most plasma samples (S2 and S3 Figs). Higher cleaning rates were observed in cfDNA negative controls (~50%) and in three cfRNA plasma samples (~25%), that may reflect lower sample input or quality. Overall, no differences in QC processing were observed between bNAb producers and non-producers.

The remaining reads were first aligned to the human genome (GRCh38, Ensembl) using bowtie2, retaining unmapped reads, which were subsequently passed through a second homology filtering by BLASTN against the NCBI nt database restricted to taxid 9606, to obtain non-human reads for microbial and viral analysis. Taxonomic classification of non-human reads was performed by BLASTN (NCBI BLAST+ v2.8) against the NCBI nt database, using curated taxid lists limited to viral, bacterial and fungal references [66,67]. BLAST hits were filtered by requiring ≥80% nucleotide identity and ≥80% per-read query coverage, also de-duplicating counts to the best-supported accession to avoid counting the same read twice across hits. Subsequently, per-accession counts and coverage profiles were generated and aggregated at taxonomic ranks for downstream analysis.

For cfRNA, reads were additionally aligned to the human genome using STAR v.25.4a [71] and duplicate-filtered using Picard v2.18.2,and gene abundance was quantified using HTSeq [72]. Downstream analysis of gene abundance data is detailed in the following section.

For HIV-specific analysis, non-human cfRNA reads were re-aligned to the HIV-1 group M consensus reference (CON-C.fa) using bowtie2. Consensus sequences were generated with samtools mpileup and bcftools consensus [73], followed by multiple sequence alignment using MAFFT v7 [74]. A maximum-likelihood phylogenetic tree was constructed using IQ-TREE [75], and visualized with Bio.Phylo [76], with metadata-based label coloring and shaded backgrounds highlighting predefined sample clusters. Previously published HIV reference sequences from the same CAPRISA participants were obtained from the Los Alamos National Laboratory (LANL) HIV Sequence Database (S2 Table). Sample identity was primarily validated by assessing HIV sequence consistency across all the three time-points per participant (Figs 2 and S3). As an orthogonal check, we performed HLA-B typing on cfRNA using seq2HLA (v2.3, 4-digit resolution) [77], which confirmed exclusion of one sample (CAP200, time point 1), where a mismatch was detected in both the HIV sequence analysis and HLA-B typing.

To control for potential environmental contamination in both cfDNA and cfRNA data, six negative controls were used to create a database of background environmental taxa that were blacklisted in downstream analysis. This filtering preserved 72% of microbial reads in cfRNA and 17% in cfDNA. In addition, non-template control barcodes (n = 8) were spiked into the cfRNA libraries to estimate sample cross-contamination, which was found to be < 0.5% and compatible with index-hopping during sequencing. This threshold was used to define the detection limit for each individual microbial taxon.

### Differential gene abundance analysis

Differential Gene Abundance Analysis was performed using edgeR with a generalized linear model and quasi-likelihood F-tests. Site of sample collection was included as a covariate in the model to account for potential confounding effects. All p-values were corrected for multiple hypothesis testing using the Benjamini-Hochberg method, and results are reported as false discovery rate (FDR). For downstream analysis, we selected genes with $FDR < 0.05$ and absolute fold change $> 1$. Associations between gene abundance and clinical metadata (e.g., viral load, CD4 + counts) were measured using nonparametric Spearman correlation unless otherwise stated (SciPy package, Python). Differential abundance of microbial taxa was also assessed in edgeR using generalized linear models, with the same thresholds and covariate adjustments. Overall, no significant site-associated differences were observed in either the transcriptomic or microbial enrichment analyses.

### Pseudotime analysis

Pseudotime analysis was used as an alternative approach to investigate factors associated with the development of neutralization breadth, given the limited number of longitudinal samples and heterogeneity in bNAb development across individuals. Samples were ordered according to the number of pseudoviruses neutralized by each plasma sample, which was used as a proxy for progression along the bNAb development continuum. This trajectory-inference approach allows us to account for differences in the pace of bNAb development across individuals. Spearman's rank correlation analyses were then performed against this pseudotime variable to evaluate changes both between bNAb producers and non-producers and along the continuous trajectory of breadth progression.

### GBV-C analysis

GBV-C reads were normalized to the total number of non-human reads per sample and scaled by a factor of 1,000 (GBV-C reads per 1,000 non-human reads), analogous to the normalization used for cfHIV. Differences in GBV-C abundance between bNAb producers and non-producers were assessed using a logistic mixed-effects model (statsmodels package, Python), where we model GBV-C abundance as a binary outcome and included participant as a random effect to account for repeated sampling across time points.

## Supporting information

**S1 Fig. Clinical information for blood samples.** Longitudinal trajectories for CD4 (orange) and viral load (purple). Diamonds indicate timepoints sequenced in this study. Grey bars indicate values for CD4 count $< 200$.
(TIF)

**S2 Fig. Sequencing quality control metrics per sample.** A) Number of reads (2x75 bp) sequenced in each sample and negative controls. Two cfDNA samples failed extraction/library preparation. B) Percentage of reads cleaned in QC steps (low quality or that align to the UniVec Core database). Percentages are generally very low, except in cfDNA negative controls, which had a high proportion of primer/library-derived sequences. C) Percentage of reads mapping to the human genome. D) Total number of non-human reads, typically more than a million for the cfRNA samples and around 10 thousand for the cfDNA samples. Samples are ordered by time within each participant.
(TIF)

**S3 Fig. Sequencing quality control metrics summarized by study group.** A) Total sequencing depth per sample (raw read counts). B) Read counts retained after QC cleaning steps (see Methods) C) Percentage of reads removed during cleaning D) Percentage of human-mapped reads after cleaning. Metrics are shown for cfRNA (left) and cfDNA (right) libraries across bNAb producers, non-producers, and negative controls. Data are shown as mean $\pm$ SEM.
(TIF)

**S4 Fig. Cell-type deconvolution analysis of cfRNA.** (a) Relative cell-type contributions to the cfRNA composition for each sample. Samples are grouped by participant and study group (bNAb producer or control), and less abundant cell types are aggregated into 'Other'. (b) Box plots showing the relative abundance of each cell type in bNAb producers and controls. Statistical significance was assessed using a Mann–Whitney U test, with p-values adjusted for multiple hypothesis testing using Benjamini–Hochberg.
(TIF)

**S5 Fig. HIV genotypes, coverage and mutations.** A) Heatmap of coverage across the genome in each sample. The four low abundance samples are marked with black squares on the left hand panel. B) Total number of read fragments aligning to HIV in each sample. C) Phylogenetic tree of HIV consensus genotypes for each sample obtained in this study together with 200 other genotypes of HIV-1 group M, subtype C from South Africa. D) Locations of SNVs local to each participant across the three time points. Most cluster within the *env* gene. E) Mutation rate observed in the samples using differences in the consensus genotypes within participants.
(TIF)

**S6 Fig. Fragment length and coverage of cfHIV reads.** A) Distribution of paired-end genomic spans for reads mapping to the HIV genome, showing a predominance of short fragments (200–600 bp). B) Mean paired-end genomic span as a function of genomic start position, showing no systematic trends that could indicate priming from long RNA templates. (c) Genomic coverage profile across the HIV genome, shown alongside the reference organization of the HIV genome.
(TIF)

**S7 Fig. Subsampling analysis of HIV genome coverage from cfRNA.** A) Fraction of the HIV genome covered above increasing minimum depth thresholds (1X-50X) at different subsampling levels of sample CAP261. B) Number of HIV-mapped reads versus mean coverage depth across the HIV genome, showing a linear relationship. C) Genome coverage achieved for each sample in the cohort at ≥10X (left) and ≥50X (right) depth thresholds.
(TIF)

**S8 Fig. Correlations of genes with different disease measurements.** A) Distributions of Spearman correlations of all genes with either cfHIV counts (top, blue), CD4 counts (middle, orange), or viral load (bottom, purple). Gray bands indicate regions of significant correlation as determined by the Spearman test on the data. The small ticks (rugplot) at the bottom of each plot indicate known HIV genes. B) Tables of the number of genes strongly correlated/anticorrelated between genes and measurements (as in A). C) Scatter plots of cfHIV abundance vs gene expression (z-score) for the 14 significantly correlated known HIV-associated genes. Besides HLA-A, none of these show strong differences between the two study groups.
(TIF)

**S9 Fig. Enriched gene pathways in bNAb producers.** A) Top gene ontologies (molecular function, biological process) and pathways enriched in genes that are elevated in bNAb producers. Numbers indicate the number of selected genes compared with the total number assigned to that category. B) Similar to A, but for the genes with decreased abundance in bNAb producers. Note that scores are much lower for decreased genes than for elevated genes and no clear enriched groups of genes are present in decreased genes. C) Distribution of genes correlated with breadth in bNAb producers (upper, red) and controls (lower, blue). D) Tables of the number of genes meeting correlation thresholds in (C).
(TIF)

**S10 Fig. Background microbial sequences in negative controls.** Volcano plot showing differential abundance of microbial genera between plasma samples and negative controls (NC) (FDR<0.05, Benjamini–Hochberg). Negative log2 fold-change values indicate genera enriched in NC, which were used to define background levels. Several NC-enriched

taxa showed large effect sizes ($|\log_2 FC| > 4$, $FDR < 10^{-8}$), primarily comprising environmental taxa previously reported in DNA extraction kits and reagents in low-biomass sequencing.
(TIF)

**S11 Fig. Phylogenetic tree of microbial sequences from circulating nucleic acids in HIV infection (all participants).** A) Phylogenetic tree of cfDNA-derived microbiome reads pooled across all participants (PLWH who do and do not develop bNAbs). The main branches represent archaea, bacteria and viruses (from left to right); representative viruses including Torque Teno Viruses (TTV), Epstein-Barr virus (EBV) and cytomegalovirus (CMV) are highlighted. B) Phylogenetic tree of cfRNA-derived microbiome reads pooled across all participants. The main branches represent archaea, bacteria and viruses (from left to right). Within the viral branch, HIV and GB virus C are the main contributors to the virome.
(TIF)

**S12 Fig. Subset of microbial genera enriched in bNAb producers.** Boxplots showing normalized abundances of selected microbial genera across negative controls (NC), bNAb non-producers, and bNAb producers. Values are shown as log2-transformed abundances relative to the median abundance in negative controls.
(TIF)

**S13 Fig. Pseudo-temporal dynamics in differentially abundant genera between bNAb producers and non-producers.** Scatter plots of the abundance (z-score) vs breadth production (%) for the 22 genera found to be differentially abundant between case (bNAbs) and control (no bNAbs). No apparent time differences found.
(TIF)

**S14 Fig. Coverage profiles of HCV and GBV-C genomes across samples.** A) Read coverage across the HCV genome. B) Read coverage across the GBV-C genome. Coverage for individual samples is shown in red (HCV) and blue (GBV-C); mean coverage is shown in black.
(TIFF)

**S1 Table. Neutralization data for the panel of 18 Env-pseudotyped HIV viruses.**
(XLSX)

**S2 Table. Participant codes and corresponding accession numbers in the HIV database (https://www.hiv.lanl.gov/content/index) for the reference FASTA sequences included in this study.**
(XLSX)

## Acknowledgments

Finally, we are indebted to the study participants for their generous support of scientific research.

## Author contributions

**Conceptualization:** Penny L Moore, Joan Camunas-Soler, Stephen R Quake.

**Data curation:** Mark Kowarsky, Mercedes Dalman, Jennifer Okamoto.

**Formal analysis:** Mark Kowarsky, Mercedes Dalman, Yike Xie, Joan Camunas-Soler.

**Funding acquisition:** Penny L Moore, Joan Camuñas-Soler, Stephen R Quake.

**Investigation:** Mark Kowarsky, Mercedes Dalman, Mira N Moufarrej, Jennifer Okamoto, Joan Camuñas-Soler.

**Methodology:** Mark Kowarsky, Mira N Moufarrej, Norma F Neff, Stephen R. Quake.

**Project administration:** Joan Camunas-Soler.

**Resources:** Norma F Neff, Salim S Abdool Karim, Nigel Garrett, Penny L. Moore.

**Supervision:** Penny L Moore, Joan Camunas-Soler, Stephen R Quake.

**Validation:** Mercedes Dalman, Yike Xie.

**Visualization:** Mark Kowarsky, Mercedes Dalman.

**Writing – original draft:** Mark Kowarsky, Joan Camunas-Soler.

**Writing – review & editing:** Mark Kowarsky, Mercedes Dalman, Mira N Moufarrej, Jennifer Okamoto, Yike Xie, Norma F. Neff, Salim S Abdool Karim, Nigel Garrett, Penny L Moore, Joan Camunas-Soler, Stephen R Quake.

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
