## [Decision Letter · Decision Letter 0]

10 Dec 2025

Cell-free RNA reveals host and microbial correlates of broadly neutralizing antibody development against HIV

PLOS Pathogens

Dear Dr. Camunas Soler,

Thank you for submitting your manuscript to PLOS Pathogens. After careful consideration, we feel that it has merit but does not fully meet PLOS Pathogens's publication criteria as it currently stands. Therefore, we invite you to submit a revised version of the manuscript that addresses the points raised during the review process.

We look forward to receiving your revised manuscript.

Kind regards,

Vijayakumar Velu, ph.D.

Academic Editor

PLOS Pathogens

Susan Ross

Section Editor

PLOS Pathogens

Editor-in-Chief

PLOS Pathogens

orcid.org/0000-0003-2946-9497

Editor-in-Chief

PLOS Pathogens

orcid.org/0000-0002-7699-2064

**Journal Requirements:**

4) In the online submission form, you indicated that "The datasets generated and analyzed in the study are available in the NCBI Gene Expression Omnibus (GEO) and Sequence Read Archive (SRA) and can be accessed upon request". All PLOS journals now require all data underlying the findings described in their manuscript to be freely available to other researchers, either

1. In a public repository

2. Within the manuscript itself

3. Uploaded as supplementary information.

**Reviewers' Comments:**

Reviewer's Responses to Questions

**Part I - Summary**

Reviewer #1: This is a pilot study of the use of cell free RNA and DNA measurements from a plasma repository of the extraordinary CAPRISA cohort. This is an admirable attempt to bring new technology to an important problem, that of defining the mechanisms of bnAb induction in PLWH. The use of statistics throughout is not consistent for each set of data.

Reviewer #2: Kowarsky et al. sampled cell-free nucleic acids longitudinally from HIV cases before antiviral therapy with the goal of finding differences between people who (go on to) develop broadly neutralising antibodies and people who do not. The hope is to find actionable environmental factors that can be tinkered with for vaccine design. Overall, the find multiple associations between HIV, host immune, and commensal or co-infecting microbial sectors. Perhaps the most interesting discovery is that during the first year after (presumed) infection, patients that will later develop broad neutralisation already show a stronger MHC-I genetic signature in their cell-free nucleic acids. They also find that bNAb producers tend to have higher relative abundance of GBV-C reads, potentially implicating an “immune priming” mechanism for early response to HIV.

In general, the study appears to have been conducted thoroughly and the cohort is really interesting for vaccine design purposes. Although only one main experimental method was used throughout, the computational analyses are diverse and informative. My impression is that, although already a useful reference as is, this manuscript could improve significantly by providing a few, relatively straightforward additions to clarify both technical and biomedical aspects of the findings. See below for specific comments.

**Part II – Major Issues: Key Experiments Required for Acceptance**

Reviewer #1: 1. Success in cfRNA, cfDNA interpretation is control for RBC hemolysis and for release of platelet RNA. How were platelet and RBC contamination controlled for in this study? what percentate of RNA was from platelet or RBCs? In a paper by the senior author’s group (PMID: 35132263) 60% of cfRNA was platelet + RBC derived.

2. That this study did not find an association between high viral load and CD4 counts with bnAb induction compared to other studies may only reflect the small sample size in this study. The authors did note that this was a pilot study and conclusions are limited.

3. In Figure 1A the difference in those with bnAbs and those with fewer bnAbs is not well demarcated. In 1A the lowest breadth % of the bnAb group and the highest breadth % of the low neutralizing group overlap at 3 years of follow up. That is unlike other such studies, as here, there is no clean break between the groups.

4. In Figure 1B, it is unclear what the boxes mean, and what log 10 affinity means in the context of pseudovirus neutralization assays. Does affinity correlate with EC50 or EC80?

5. Figure 3A, it is unclear what test was used for the p values shown.

6. Figure 4, correlations are presented but no tests are indicated how they were obtained.

7. In Figure 5 there does not appear to be statistically significant differences in GBV-C levels in the two groups. No statistical methods noted to analyze the data.

8. In Figure S2 there appear to be as many sequences in the water control as in the samples. Is this usual? How are relevant cf genetic sequences identified with such high background of contamination?

9. Also in Figure 2 why were so many contaminating sequences “cleaned” from the bnAb group but not in the control group?

10. In Figure S3, was the RNA outside of cells and outside of virions or does this sequencing simply represent virion-associated RNA? Were the samples pelleted by ultracentrifugation to remove intact virions?

11. Figure S6 is hard to understand. Are these data from only the PLWH who made high bnAbs? Or both groups?

12. Figure S7. What are pseudo-temporal dynamics? Definition would help the reader.

13. Several uses of the word “significantly” (as in line 193) with no reference to statistical test or giving a P value.

14. Similarly the word “correlated” or “anti-correlated” was used (as in lines 209-210) with no correlation coefficient or statistical test described.

Reviewer #2: • Data should be on GEO with identifier, and code should be available e.g. on GitHub.

• Although the authors demonstrate that they can reconstruct HIV genomes using cell-free nucleic acids, it is unclear to me how many reads they need to achieve what kind of assembly quality. Some downsampling to get a sense of how many cfRNA reads you would need to assemble an HIV genome to what degree/quality would really help future researchers tailor their experimental design.

• On line 170, the authors state that CCL5 etc. are responses to HIV infection. Because these genes have very broad immune functions, the authors should explore the immunology literature to clarify whether these genes are responses not just shared between HIV bNAb groups, but also with patients infected by other viruses or even bacteria. The authors should comment on this issue based on a literature search and potentially cross-examine third-party data from previous publications to check the specificity of this signature.

• The authors mention GBV-C but do not elaborate whether other flaviruses were also found. HCV-HIV coinfection is not that uncommon and would cause a quite different immunological response, therefore the authors should clarify whether they specifically see Hepatitis C reads in their samples.

• Did the pseudovirus neutralisation assay in Figure 1B for non-bNAb producers match the strain that was found in their cfRNA reads? In other words, can their serum at least neutralise narrowly their own strain?

**Part III – Minor Issues: Editorial and Data Presentation Modifications**

Reviewer #1: minor comments:

15. The participants in the CAPRISA cohort study are referred to as either “donors” or “patients” in the paper (line 235). Perhaps the generic term “participants” might be more consistent.

16. Vague phrases throughout. Line 43, “activation that involves MHC class I antigen presentation”; line 62 “changes in commensal microbiota and increased GB virus C co-infection” (increased levels or increased number of participants?); GB virus C is mentioned several times in the paper before it is defined in line 251. It should be defined early on in the paper.

Reviewer #2: The first sentence of the abstract is not strictly true: a narrow response is ok to prevent viral infections if the strain matches. Hence the yellow fever vaccine. Rephrasing is needed.

Not sure CCL5 etc. are “covariates of HIV infection progression” as much as they are proxies of high viral titers in the periphery. The two things are not quite the same as progression is clinically defined based on CD4 counts rather than viral load.

The authors dutifully cite many of the software packages used for the analyses but miss others. For instance, they presumably used HTSeq 2.0 but fail to cite it. They should amend this inconsistency.

The discussion section about lymph nodes is highly speculative and should be tempered to avoid giving the impression that this manuscript provides hard evidence that specific signature found herein can be attributed unequivocally to lymph node processes.

In addition to the above, I have a specific suggestion that, albeit not 100% necessary for the manuscript, would elevate the conclusions of the paper and potentially shed light on new biology:

• The correlation between higher MHC-I production transcripts and the potential to later develop bNAbs is very interesting. MHC-I presentation is usually seen as an immune pioneering process that then activates cytotoxic T cells (CTLs), thereby contributing to the early stages of the immune activation cascade. However, some viral proteins (e.g. HIV Nef https://www.nature.com/articles/nri2575) can downregulate MHC-I presentation as an escape mechanism from CTLs. These data suggests that perhaps in bNAb producers, early inhibition of MHC-I by viral factors is less effective, buying enough time for the immune system to take the bNAb route. If so, anti-Nef drugs or a Nef-mutant viral strain would be an interesting direction for vaccine efforts. The authors could test the hypothesis that patients with stronger MHC-I production early on (before returning to baseline) also have relatively fewer Nef reads (compared to the total of HIV reads) and/or mutations in Nef that might decrease its effectiveness (i.e. nonsynonymous mutations in relatively conserved residues of the protein). Potentially, the authors could also take the Nef sequences from samples with high vs low MHC-I production early on and predict structures using AlphaFold 3 (online server) to test the hypothesis that these specific Nef strains might differ structurally from the ones derived from non bNAbs producers.

PLOS authors have the option to publish the peer review history of their article (what does this mean? ). If published, this will include your full peer review and any attached files.

**Do you want your identity to be public for this peer review?** For information about this choice, including consent withdrawal, please see our Privacy Policy .

Reviewer #1: No

Reviewer #2: **Yes:** Fabio Zanini

**Figure resubmission:**

**Reproducibility:**



---

## [Editor Report · Decision Letter 1]

5 Mar 2026

Dear Dr. Camunas Soler,

We are pleased to inform you that your manuscript 'Cell-free RNA reveals host and microbial correlates of broadly neutralizing antibody development against HIV' has been provisionally accepted for publication in PLOS Pathogens.

Best regards,

Vijayakumar Velu, ph.D.

Academic Editor

PLOS Pathogens

Susan Ross

Section Editor

PLOS Pathogens

Sumita Bhaduri-McIntosh

Editor-in-Chief

PLOS Pathogens

orcid.org/0000-0003-2946-9497

Michael Malim

Editor-in-Chief

PLOS Pathogens

orcid.org/0000-0002-7699-2064
---

## [Editor Report · Acceptance letter]

Dear Dr. Camunas Soler,

We are delighted to inform you that your manuscript, "Cell-free RNA reveals host and microbial correlates of broadly neutralizing antibody development against HIV," has been formally accepted for publication in PLOS Pathogens.

Best regards,

Sumita Bhaduri-McIntosh

Editor-in-Chief

PLOS Pathogens

orcid.org/0000-0003-2946-9497

Michael Malim

Editor-in-Chief

PLOS Pathogens

orcid.org/0000-0002-7699-2064